

# Improving size selectivity of round pot for *Charybdis japonica* by configuring escape vents in the Yellow Sea, China

Mengjie Yu, Liyou Zhang, Changdong Liu and Yanli Tang

Fisheries College, Ocean University of China, Qingdao, China

## ABSTRACT

Sustainable development of the important economic species, Asian paddle crab (*Charybdis japonica*), has attracted attention in the coastal waters of the Yellow Sea, China. The commonly used round pots are almost nonselective, resulting in severe bycatch of juveniles. In this study, we explored a method to improve the size selectivity for *C. japonica* by mounting escape vents on the side panels of each pot. The selectivity of pots with escape vent sizes of 70 mm × 20 mm, 70 mm × 25 mm, 70 mm × 30 mm, and 70 mm × 35 mm was tested using a catch comparison method. The estimated minimum landing size (MLS) of carapace height (27 mm), according to the regulated MLS of carapace length (50 mm), was used as a reference point to explain the results. Significant increases in the size of crabs caught by pots were found with the enlargement of escape vent size (Kruskal–Wallis test, $P < 0.01$). The pots with 70 mm × 20 mm, 70 mm × 25 mm and 70 mm × 30 mm escape vents released nearly 50%, 75% and 95% of undersized individuals, respectively, and these three types of pots retained approximately 90% of legal-sized individuals compared with the control pots without escape vents. The pots with 70 mm × 35 mm escape vents released nearly all undersized individuals, but they also released most legal-sized individuals. Pots with an escape vent size of 70 mm × 30 mm were recommended for the sustainable development of *C. japonica* in the Yellow Sea of China. The results of this study reiterate the importance of carapace height for determining the size selectivity, which can serve as a reference to formulate management regulations in the coastal waters of the Yellow Sea, China.

## INTRODUCTION

Crustacean fisheries represent an important part of marine commercial catches in China, recently comprising approximately 19.2% of the total landing (approx. ten million tons) (*Fisheries Administration Bureau, MARA, PRC, 2020*). Crabs are among the most important crustacean catches and can be caught by many types of fishing gear, including trawls, gillnets, set nets, hoop nets, traps, and pots. Pots are well known for their high species selection and have become the main fishing gear for crabs (*Yu et al., 2003*; *Vazquez Archdale & Kuwahara, 2005*; *Song et al., 2006*; *Zhang et al., 2020*).

Corresponding authors
Changdong Liu,
changdong@ouc.edu.cn
Yanli Tang, tangyanli@ouc.edu.cn

Steel-framed round pots (hereafter 'round pots'), with the attributes of low cost and labor requirements, habitat friendliness and high catch efficiency, are widely used in the coastal waters of the Yellow Sea, China. Compared with commonly used fishing gear (*e.g.*, trawls), round pots can be deployed in natural rocky/artificial reef areas. Round pots are also portable and easy to place aboard, enabling fishermen to work more efficiently. Furthermore, round pots are less prone to being wrecked and have a longer effective working life. Crabs caught by round pots have high quality, as most of them are alive and undamaged when landing, achieving a higher price in the market.

The Asian paddle crab, *Charybdis japonica*, belonging to settlement crustaceans, is widely distributed in the muddy, sandy and rocky/reef areas of the coastal waters of China, Korea, Japan and Southeast Asia at depths of 9–45 m (*Vazquez Archdale & Kuwahara, 2005*; *Zhang et al., 2016*). Because of its high nutritional and economic value, *C. japonica* is an important commercial species and is one of the most important targeted species for round pots in the Yellow Sea of China. This species matures at a carapace length (*CL*) of 50 mm, so a minimum landing size (MLS) of 50 mm *CL* was formulated according to decree No. 34 of the Chinese Ministry of Agriculture. The MLS of carapace height (*CH*) was estimated to be 27 mm based on the relationship between *CL* and *CH* for *C. japonica* (see "Results").

Despite the good attributes mentioned above, the currently used round pots in the Yellow Sea of China present poor selectivity for *C. japonica* due to their small mesh size (25 mm). In recent decades, many researchers have sought to improve the size selectivity of pots for different species by mounting escape vents/gaps, increasing mesh size, modifying mesh shape, and adjusting pot shape or entrance design (*Guillory & Hein, 1998*; *Vazquez Archdale & Kuwahara, 2005*; *Vazquez Archdale et al., 2007*; *Boutson et al., 2009*; *Broadhurst, Millar & Hughes, 2017*, *2018*; *Broadhurst et al., 2019*). Installing escape vents or gaps has been proven to be a more convenient and effective measure to improve selectivity and optimize catching efficiency in different pot fisheries (*e.g.*, *Brown, 1982*; *Tallack, 2007*; *Boutson et al., 2009*; *Rotherham et al., 2013*). Meanwhile, this method is more precise for size selection than enlargement of mesh size (*Nishiuchi, 2001*; *Winger & Walsh, 2007*).

According to behavioral observations in the laboratory, crabs seek to escape from the vents by side-crawling, so the size selectivity of *C. japonica* for *CL* depends on the vent length, and *CH* depends on the vent width (*Boutson et al., 2009*). Although several studies have investigated the effect of escape vent length on the size selectivity of crabs (*Zhang & Zhang, 2013*; *Zhang et al., 2016*; *Broadhurst, Millar & Hughes, 2018*), comprehensive studies on the size selectivity of escape vents for *C. japonica* in the Yellow Sea of China have not been performed.

In this study, we improved the size selectivity of round pots for *C. japonica* by mounting escape vents from a small-scale fishery in the coastal waters of the Yellow Sea, China. We tested the size selectivity of four different escape vent width sizes (20, 25, 30, and 35 mm) with a fixed escape vent length of 70 mm using a catch comparison method. Five selective models were tested, and the best model was selected. The main objective of this study was to quantify the size selectivity of escape vents for *C. japonica* using selective

models and exploitation pattern indicators and then recommend an optimal escape vent size for this species considering local fishermen's profit and resource sustainability.

## MATERIALS AND METHODS

### Gear modification

The conventional round pot, which is commonly used in the coastal waters of the Yellow Sea, China, is cylindrical in shape with a height of 25 cm and diameter of 55 cm. Each pot consists of six components, including a rigid steel frame, cover net (PE-36tex × 4 × 3; R12tex S/Z), funnel net (PE-36tex × 2 × 3; R6tex S/Z), bait basket, mouth rope (PE-36tex × 25 × 3; R75tex S/Z) and hook. The transversal and longitudinal mesh numbers of the cover net are 31 and 67, respectively, and those of the funnel net are 14 and 36, respectively. The mesh sizes (±SD) of the cover net and funnel net are 25 mm (±0.09) and 20 mm (±0.10), respectively.

The round pot has six sides in total, three of which have funnel-shaped entrances. Laboratory observations showed that the bottom of the side panel was the best position for crabs to escape. Therefore, three escape vents cut from a board (thickness: 5 mm) made from PMMA (polymethyl methacrylate) were mounted on the bottom of the three sides and fixed by nylon cable ties (HDS-5 × 350 mm, quantity 250 pcs, tension 22 kg/50 lbs, bundle diameter 3–90 mm) (Fig. 1). We modified the round pot with four different escape vent sizes (length × width: 70 × 20 mm, 70 × 25 mm, 70 × 30 mm, and 70 × 35 mm). The length of each escape vent was set uniformly as 70 mm to ensure that *C. japonica* was not affected by *CL* for escapement. The concrete pot modification is shown in Fig. 1.

### Sea trials

The fishing experiment was conducted in the coastal waters of the Yellow Sea, China (35°15′0″–35°16′30″ N and 119°28′30″–119°30′0″ E) from August 19 to September 7, 2020. The study area is a traditional fishing ground for *C. japonica* and was deployed artificial reefs from 2005 to 2014. The crab populations are more abundant in this area than those in the natural sea area, making us to catch enough crabs to conduct the selectivity analysis. The sampling sites were selected based on the recommendation of an experienced local fisherman (Fig. 2).

In the sea trials, we used the catch comparison method, and a total of 50 pots were deployed, including ten conventional pots as the control group and 40 modified pots (ten pots for each type) as the test group. Control pots with 25 mm mesh size were assumed to be non-selective, so their catch is representative of the stock in the fishing ground. Ten pots (two control pots and two pots with 70 mm × 20 mm, 70 mm × 25 mm, 70 mm × 30 mm, and 70 mm × 35 mm escape vents) were connected into a string with a random sequence. A total of five strings were used during the experiment, and each end of the string was connected with buoys and anchors (weighing 15 kg per anchor) (Fig. 1). A local vessel, the "Lurigangyu 77369" (LOA 8.3 m, width 3.2 m, height 0.85 m, weight 4.0 GT, power 16.2 Kw), with a vessel speed of approximately 3.0 knots and

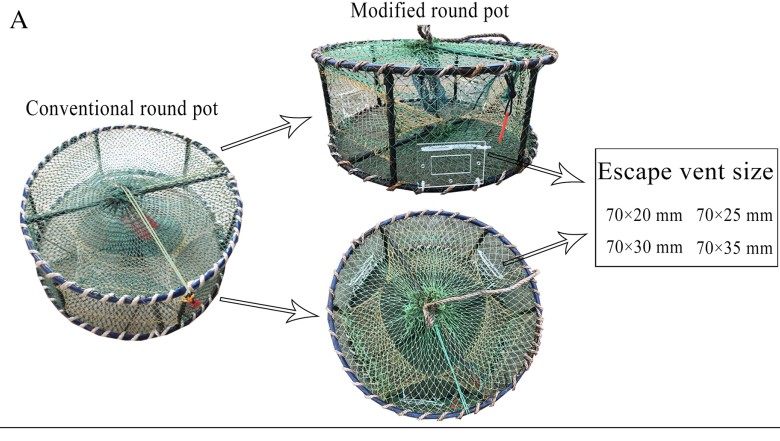

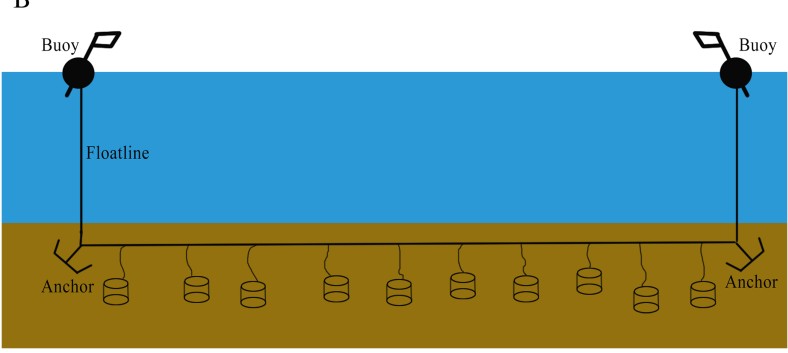

**Figure 1 Experimental gear configuration and operation.** (A) Illustration of the modified round pot and escape vent size (70 mm × 20 mm, 70 mm × 25 mm, 70 mm × 30 mm, and 70 mm × 35 mm). (B) Deployment of pots in a single string. Each string had ten pots, and a total of five strings were used during the sea trials. The pots were deployed 50 m apart with two anchors each weighing 15 kg at the ends.

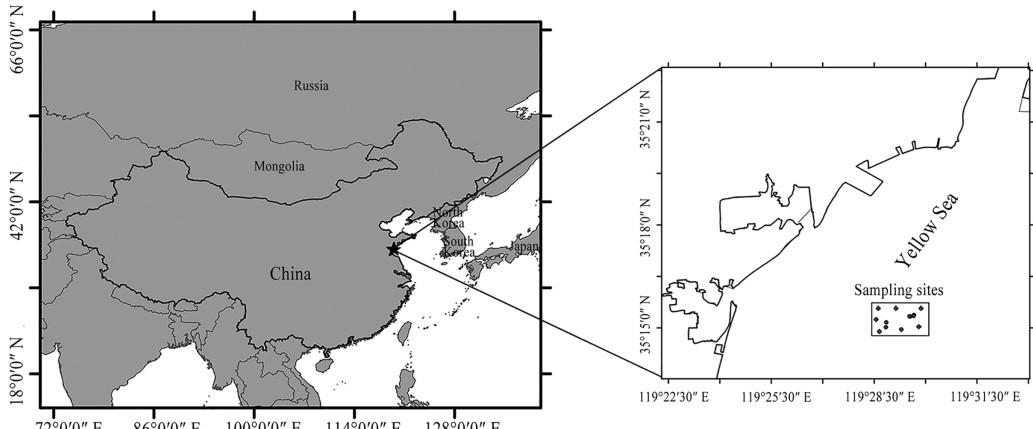

**Figure 2 The location of the sampling sites in the Yellow Sea of China.** The substrate type is a mixture of mud, sand, and rock.

operated by three experienced local fishermen, was used to deploy and retrieve pots. Pots were deployed 50 m apart to ensure independence at a water depth of 12–14 m and baited with *Mytilus edulis*, approximately 150 g per pot. All pots were hauled up in the

morning after approximately 2 days of soaking time, catches were collected, and then the pots were replaced in their original location.

All crabs captured were measured for *CL* (the distance from the frontal notch to the posterior margin of the carapace), carapace width (*CW*, defined as the distance between the ninth anterolateral spines), and *CH* (measured from the base of the second sternal segment to the highest part of the gastric region) to the nearest mm and for weight (*W*) to the nearest 0.01 g.

## Model for size selection

The SELECT (Share Each Length's Catch Total) method was used to estimate the size selectivity by analyzing the proportion of individuals caught by each type of pot (for each *CH* class). The retention rate ($\Phi_{jCH}$) of individuals in class *CH* caught by pots with *j*-sized escape vents *versus* the total catch can be expressed as follows:

$$\Phi_{jCH} = \frac{N_{jCH}}{\sum_j^n N_{jCH}} = \frac{p_j s_j(CH, v)}{\sum_j^n p_j s_j(CH, v)} \tag{1}$$

where $N_{jCH}$ is the number of individuals in the *CH* class caught by pots with *j*-sized escape vents, $p_j$ is the relative fishing intensity (defined as the probability that an individual entered the round pot with a *j*-sized escape vent, given that it entered the combined (test and control) gears), $\sum_j p_j = 1$, and $s_j(CH, v)$ is the retention probability with the unknown selectivity parameter vector *v* (*Wileman et al., 1996*; *Yang, Tang & Liang, 2011*; *Tang et al., 2019*).

Data were pooled from different hauls to estimate average size selection over hauls, $s_{av}(CH, v)$. Five different models were chosen as candidates to describe $s_{av}(CH, v)$: *Logit*, *Probit*, *Log-log*, *Clog-log*, and *Richards*. The vector *v* consists of two selectivity parameters, CH50 (*CH* of *C. japonica* with 50% probability of being retained) and SR (difference in *CH* of *C. japonica* with 25% and 75% probability of being retained). The *Richard* model needs one additional parameter ($1/\delta$) to describe the asymmetry of the size selection curve. Detailed information about these models can be found in *Wileman et al. (1996)*.

To estimate the parameter values that make the experimental data most likely to be observed, selectivity parameters and the relative fishing intensity $p_j$ were determined by maximizing the log-likelihood function as follows:

$$\sum_j \sum_{CH} N_{jCH} ln \Phi_j(CH) \tag{2}$$

where the two sums are for pot deployments conducted with the *j*-sized escape vent and *CH* classes, respectively.

The capacity of each model to fit the data was inspected based on the goodness-of-fit (*P*-value) as described by *Wileman et al. (1996)*. The model performances were compared based on Akaike information criterion (AIC) values, and the best model was selected with the lowest AIC value (*Akaike, 1974*). We used the best model to model the selectivity curves.
Once the size selection model was identified, a double bootstrap method was used to estimate the 95% confidence interval ($CI$) for the size selection curves and the corresponding parameters. This method is identical to the one used in *Millar (1993)*, which takes both between-haul and within-haul variation into consideration.

To evaluate the effect of escape vent size, the differences (Delta) between $CH$-dependent retention rates were estimated as follows:

$$\delta_1 s(CH) = s_{70\times20}(CH) - s_{control}(CH); \quad \delta_2 s(CH) = s_{70\times25}(CH) - s_{70\times20}(CH);$$
$$\delta_3 s(CH) = s_{70\times30}(CH) - s_{70\times25}(CH); \quad \delta_4 s(CH) = s_{70\times35}(CH) - s_{70\times30}(CH)$$

(3)

$\delta s(CH)$ varies from −1 to 1 and for those $CH$ classes in which the $CI$ for $\delta s(CH)$ did not contain 0.0, indicating a significant difference in selectivity between the two escape vents.

All estimates were obtained using the software tool R (*R Core Team, 2018*), and 1,000 bootstrap repetitions were conducted to obtain $CIs$ based on the "boot" package.

## Estimation of exploitation pattern indicators

Exploitation pattern indicators were used to assess the capture efficiency considering the size structure of the population caught during the sea trials (*Brinkhof et al., 2020*). Because the escape vent length was fixed and the escape vent width depended on the $CH$, the relationship between the $CH$ and $CL$ of *C. japonica* was built to calculate the MLS of $CH$ based on the formulated MLS of $CL$ (50 mm). To evaluate how the escape vents would affect a specific fishery and estimate the percentages of individuals retained (in number) below and above the MLS of $CH$, three exploitation pattern indicators, $nP_-$, $nP_+$ and $nRatio$ ($nRatio = nP_-/nP_+$) (Eq. (4)), were calculated for each test pot using the individuals caught in the control pot as the baseline. These indicators are expressed as:

$$nP_- = 100 \times \frac{\sum_j \sum_{ch<MLS} nT_{jch}}{\sum_j \sum_{ch<MLS} nC_{jch}}$$

$$nP_+ = 100 \times \frac{\sum_j \sum_{ch\geq MLS} nT_{jch}}{\sum_j \sum_{ch\geq MLS} nC_{jch}}$$

$$nRatio = \frac{\sum_j \sum_{ch<MLS} nT_{jch}}{\sum_j \sum_{ch\geq MLS} nT_{jch}}$$

(4)

where the summation of $j$ is over hauls with a specific escape vent size, and $ch$ is over $CH$ classes. $nT_{jch}$ and $nC_{jch}$ represent the number of individuals of $CH$ class in haul $j$ found in the test and control pots, respectively. $nP_-$ and $nP_+$ indicate the retention efficiency of test pots for individuals below and above the MLS of $CH$. A lower value of $nP_-$ and a higher value of $nP_+$ indicate that the pots release more undersized individuals and retain more legal-sized individuals, which is beneficial to the sustainable development of fishery resources. $nRatio$ indicates the ratio of undersized and legal-sized individuals retained.

The above indicators were based on the number of individuals, but the value of catch was more related to weight. Therefore, the relationship between $CH$ and $W$ was built, and similar indicators ($wP_-$, $wP_+$, $wRatio$) based on weight were also estimated:

$$wP_- = 100 \times \frac{\sum_j \sum_{ch<MLS}\{w_{ch} \times nT_{jch}\}}{\sum_j \sum_{ch<MLS}\{w_{ch} \times nC_{jch}\}}$$

$$wP_+ = 100 \times \frac{\sum_j \sum_{ch \geq MLS}\{w_{ch} \times nT_{jch}\}}{\sum_j \sum_{ch \geq MLS}\{w_{ch} \times nC_{jch}\}}$$

$$wRatio = \frac{\sum_j \sum_{ch<MLS}\{w_{ch} \times nT_{jch}\}}{\sum_j \sum_{ch \geq MLS}\{w_{ch} \times nT_{jch}\}} \tag{5}$$

The double bootstrap method was used to estimate the Efron percentile 95% confidence limits for the indicators ($nP_-$, $nP_+$, $nRatio$, $wP_-$, $wP_+$, $wRatio$) considering both the effect of between-pot variation and the uncertainty related to within-pot variation (*Herrmann et al., 2012*; *Cheng et al., 2019*; *Kalogirou et al., 2019*).

Kruskal–Wallis test was used to examine whether there was a significant increase in the size of crabs caught by the pots with the enlargement of escape vent size. The significance of $nP_-$ and $nP_+$ were also examined using the Kruskal–Wallis test. With the comprehensive consideration of the released rate of undersized individuals and retained rate of legal-sized individuals, we recommended an optimal escape vent size for *C. japonica*.

## RESULTS

A total of 50 pots (ten for each type) were used during the 20-day sea trials, one pot was lost, and supplementation was conducted in a timely manner. None of the PMMA board was fractured or fell off. A total of 704 *C. japonica* were caught in 9 hauls, ranging in $CH$ from 9 to 45 mm and accounting for 64% and 70% of the total catch number and weight, respectively. The number of *C. japonica* in pots with different escape vent sizes (control, 70 mm × 20 mm, 70 mm × 25 mm, 70 mm × 30 mm, and 70 mm × 35 mm) was 250, 165, 126, 99, and 64, respectively. The $CH$ distribution of *C. japonica* caught in each type of pot is shown in Fig. 3. There was a significant increase in the size of retained crabs with the enlargement of escape vent size (Kruskal–Wallis test, $P < 0.01$) (Table 1). The by-catch species included *Rapana venosa*, *Palaemon ortmanni*, *Pennahia argentata*, and *Asterinidae*. The catch species composition of different pot types was similar and the catch rates of by-catch decreased with the enlargement of escape vent size (Table 1).

The five selectivity models (*Logit*, *Loglog*, *Probit*, *Cloglog*, and *Richard*) provided acceptable $P$-values ($P > 0.05$), showing that they were well fitted for the data. The traditional logistic model was selected as the best model because of its lowest AIC value, so the following results were obtained based on this model.

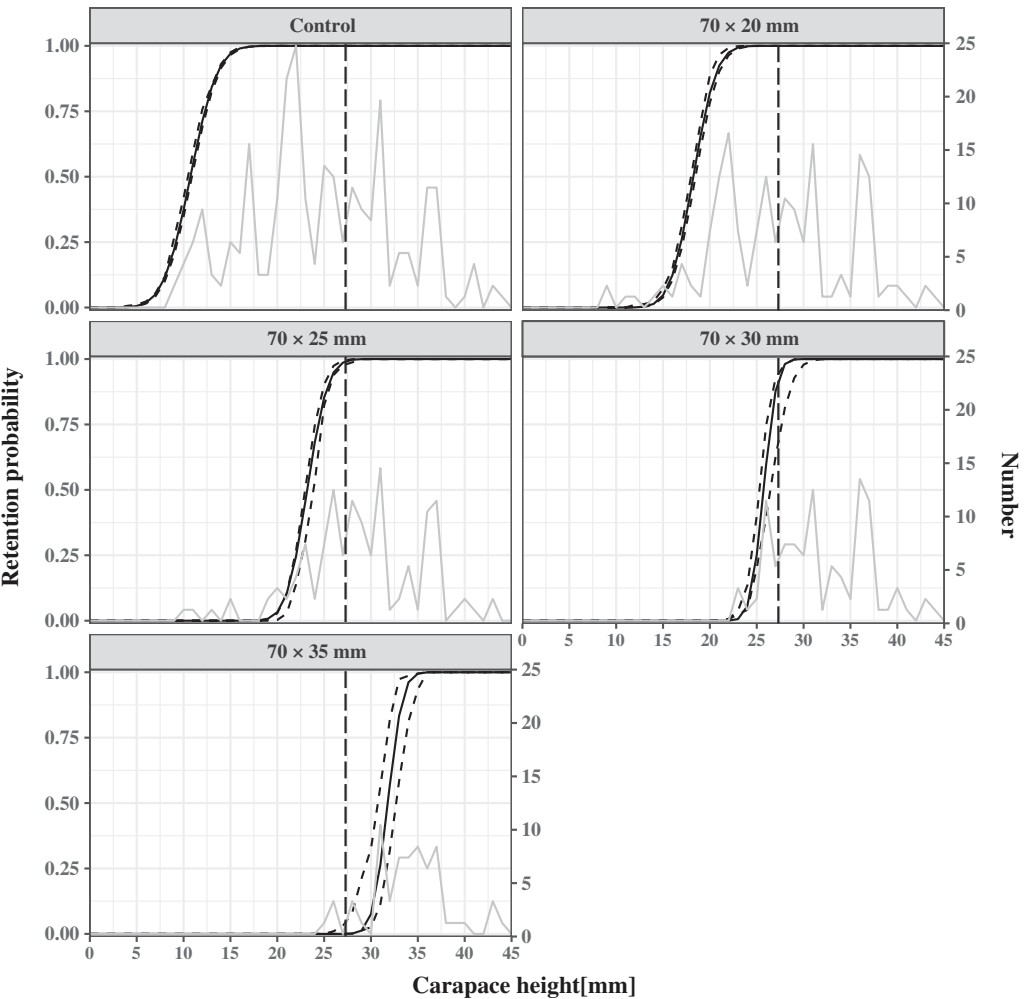

**Figure 3 Mean size selectivity curves estimated for *C. japonica* in pots with different escape vent sizes (control, 70 mm × 20 mm, 70 mm × 25 mm, 70 mm × 30 mm, and 70 mm × 35 mm).** Thick solid lines represent the mean size selectivity curves. The dashed curves represent 95% confidence intervals. The vertical dashed line indicates the MLS of carapace height (27 mm). The gray lines represent the carapace height distribution of *C. japonica* caught by each type of pot.

The estimated relationships between *CH* and *CL*, *W* and *CH* can be described as:

$$CH = 0.62CL - 3.88 \quad R^2 = 0.98$$

$$W = 1.12 \times 10^{-2} CH^{2.62} \quad R^2 = 0.89$$

The MLS of *CH* was calculated as 27 mm based on the formulated MLS of *CL* (50 mm). The CH50 was estimated at 10.73 mm (*CI*: 10.45–10.91), 18.23 mm (*CI*: 17.98–18.43), 23.19 mm (*CI*: 22.93–23.85), 25.73 mm (*CI*: 25.02–26.25), and 31.79 mm (*CI*: 30.80–32.52) for the control pot and escape vent sizes of 70 mm × 20 mm, 70 mm × 25 mm, 70 mm × 30 mm, 70 mm × 35 mm, respectively (Table 2), demonstrating that the CH50 increased with increasing escape vent size. The SR was estimated at 4.97 mm (*CI*: 4.61–5.28), 4.23 mm (*CI*: 3.93–4.54), 3.79 mm (*CI*: 3.60–4.13), 2.40 mm

**Table 1 Details of catch data from the sea trials. CH is the carapace height; SD is the standard deviation.**

| Trip | Pot type | Date | Soak time (h) | C. japonica | | | Bycatch | | | |
|---|---|---|---|---|---|---|---|---|---|---|
| | | | | Total no. caught | CH range (mm) | Average CH (±SD) | Rapana venosa | Palaemon ortmanni | Pennahia argentata | Asterinidae |
| 1 | Control | 20/08/2020 | 48 | 33 | 12–37 | 22.7 (±7.9) | 10 | 2 | 1 | 3 |
| | 70 mm × 20 mm | 20/08/2020 | 48 | 18 | 17–39 | 28.0 (±6.1) | 6 | 2 | 1 | 2 |
| | 70 mm × 25 mm | 20/08/2020 | 48 | 14 | 20–39 | 29.6 (±5.4) | 4 | 1 | 0 | 3 |
| | 70 mm × 30 mm | 20/08/2020 | 48 | 10 | 26–40 | 32.0 (±4.3) | 4 | 1 | 2 | 2 |
| | 70 mm × 35 mm | 20/08/2020 | 48 | 7 | 29–37 | 33.0 (±2.6) | 3 | 0 | 1 | 1 |
| 2 | Control | 22/08/2020 | 48 | 21 | 21–43 | 27.9 (±6.2) | 8 | 1 | 2 | 4 |
| | 70 mm × 20 mm | 22/08/2020 | 48 | 17 | 15–36 | 27.2 (±6.0) | 5 | 0 | 1 | 4 |
| | 70 mm × 25 mm | 22/08/2020 | 48 | 14 | 19–37 | 29.6 (±4.9) | 2 | 1 | 3 | 1 |
| | 70 mm × 30 mm | 22/08/2020 | 48 | 10 | 26–40 | 31.2 (±4.5) | 3 | 0 | 1 | 2 |
| | 70 mm × 35 mm | 22/08/2020 | 48 | 7 | 31–35 | 32.7 (±1.6) | 2 | 0 | 0 | 3 |
| 3 | Control | 24/08/2020 | 48 | 27 | 12–41 | 25.1 (±7.3) | 7 | 3 | 1 | 5 |
| | 70 mm × 20 mm | 24/08/2020 | 48 | 17 | 15–37 | 27.2 (±6.0) | 5 | 1 | 2 | 3 |
| | 70 mm × 25 mm | 24/08/2020 | 48 | 14 | 21–43 | 31.0 (±6.7) | 6 | 0 | 0 | 3 |
| | 70 mm × 30 mm | 24/08/2020 | 48 | 9 | 25–37 | 31.8 (±4.4) | 3 | 1 | 0 | 2 |
| | 70 mm × 35 mm | 24/08/2020 | 48 | 6 | 31–36 | 33.7 (±1.7) | 3 | 0 | 0 | 2 |
| 4 | Control | 26/08/2020 | 47 | 42 | 9–41 | 21.9 (±7.5) | 8 | 2 | 3 | 4 |
| | 70 mm × 20 mm | 26/08/2020 | 47 | 19 | 9–43 | 26.3 (±8.1) | 5 | 2 | 0 | 2 |
| | 70 mm × 25 mm | 26/08/2020 | 47 | 14 | 15–37 | 28.2 (±5.8) | 3 | 1 | 1 | 2 |
| | 70 mm × 30 mm | 26/08/2020 | 47 | 11 | 23–37 | 30.5 (±4.5) | 2 | 0 | 1 | 4 |
| | 70 mm × 35 mm | 26/08/2020 | 47 | 6 | 25–37 | 31.2 (±4.4) | 2 | 0 | 1 | 2 |
| 5 | Control | 28/08/2020 | 48 | 14 | 23–43 | 30.5 (±5.0) | 5 | 0 | 2 | 2 |
| | 70 mm × 20 mm | 28/08/2020 | 48 | 19 | 11–40 | 27.5 (±7.7) | 7 | 1 | 1 | 4 |
| | 70 mm × 25 mm | 28/08/2020 | 48 | 14 | 15–37 | 27.8 (±6.0) | 5 | 3 | 0 | 2 |
| | 70 mm × 30 mm | 28/08/2020 | 48 | 11 | 24–37 | 31.2 (±4.3) | 4 | 1 | 1 | 0 |
| | 70 mm × 35 mm | 28/08/2020 | 48 | 6 | 26–37 | 32.3 (±4.1) | 3 | 0 | 0 | 4 |
| 6 | Control | 30/08/2020 | 48 | 29 | 11–41 | 25.1 (±8.2) | 9 | 1 | 1 | 6 |
| | 70 mm × 20 mm | 30/08/2020 | 48 | 20 | 12–43 | 28.1 (±7.6) | 4 | 0 | 2 | 3 |
| | 70 mm × 25 mm | 30/08/2020 | 48 | 14 | 13–37 | 27.9 (±6.2) | 5 | 0 | 2 | 3 |
| | 70 mm × 30 mm | 30/08/2020 | 48 | 14 | 26–44 | 33.1 (±5.4) | 1 | 0 | 0 | 3 |
| | 70 mm × 35 mm | 30/08/2020 | 48 | 8 | 26–45 | 35.0 (±5.6) | 2 | 0 | 0 | 1 |
| 7 | Control | 01/09/2020 | 49 | 36 | 10–44 | 23.2 (±8.3) | 5 | 2 | 0 | 3 |
| | 70 mm × 20 mm | 01/09/2020 | 49 | 19 | 9–39 | 27.2 (±7.4) | 6 | 2 | 1 | 4 |
| | 70 mm × 25 mm | 01/09/2020 | 49 | 14 | 11–40 | 28.4 (±7.1) | 3 | 0 | 1 | 3 |
| | 70 mm × 30 mm | 01/09/2020 | 49 | 13 | 23–43 | 33.1 (±5.8) | 5 | 0 | 1 | 1 |
| | 70 mm × 35 mm | 01/09/2020 | 49 | 11 | 31–43 | 37.3 (±3.5) | 0 | 1 | 1 | 1 |
| 8 | Control | 03/09/2020 | 47 | 21 | 18–41 | 28.0 (±6.1) | 6 | 2 | 2 | 5 |
| | 70 mm × 20 mm | 03/09/2020 | 47 | 18 | 17–41 | 27.5 (±7.4) | 4 | 1 | 2 | 4 |
| | 70 mm × 25 mm | 03/09/2020 | 47 | 14 | 10–40 | 28.2 (±7.1) | 4 | 1 | 0 | 2 |
| | 70 mm × 30 mm | 03/09/2020 | 47 | 11 | 25–43 | 32.4 (±5.0) | 3 | 1 | 0 | 2 |
| | 70 mm × 35 mm | 03/09/2020 | 47 | 7 | 28–43 | 34.4 (±4.5) | 4 | 1 | 0 | 1 |

(Continued)

| Table 1 (continued) | | | | | | | | | |
|---|---|---|---|---|---|---|---|---|---|
| Trip | Pot type | Date | Soak time (h) | *C. japonica* | | | Bycatch | | | |
| | | | | Total no. caught | CH range (mm) | Average CH (±SD) | *Rapana venosa* | *Palaemon ortmanni* | *Pennahia argentata* | *Asterinidae* |
| 9 | Control | 05/09/2020 | 48 | 27 | 10–37 | 24.6 (±7.8) | 2 | 3 | 1 | 4 |
| | 70 mm × 20 mm | 05/09/2020 | 48 | 18 | 17–43 | 28.1 (±6.6) | 6 | 2 | 1 | 3 |
| | 70 mm × 25 mm | 05/09/2020 | 48 | 14 | 20–43 | 29.9 (±6.3) | 4 | 1 | 2 | 2 |
| | 70 mm × 30 mm | 05/09/2020 | 48 | 10 | 23–37 | 31.6 (±4.7) | 3 | 0 | 1 | 1 |
| | 70 mm × 35 mm | 05/09/2020 | 48 | 6 | 31–36 | 33.3 (±1.6) | 2 | 0 | 0 | 1 |
| Total | Control | | 47–49 | 250 | 9--44 | 24.7 (±7.8) | 60 | 16 | 13 | 36 |
| | 70 mm × 20 mm | | 47–49 | 165 | 9–43 | 27.4 (±7.1) | 48 | 11 | 11 | 29 |
| | 70 mm × 25 mm | | 47–49 | 126 | 10–43 | 29.0 (±6.3) | 36 | 9 | 9 | 21 |
| | 70 mm × 30 mm | | 47–49 | 99 | 23–44 | 31.9 (±4.9) | 28 | 4 | 7 | 17 |
| | 70 mm × 35 mm | | 47–49 | 64 | 25–45 | 34.0 (±4.1) | 21 | 2 | 3 | 16 |

(*CI*: 2.11–2.52), and 2.74 mm (*CI*: 2.38–3.05) for the five pot types, respectively (Table 2). The SR decreased with increasing escape vent size. Based on the estimated parameters, the mean size selectivity curves and their *CIs* are shown in Fig. 3 with MLS of *CH* (27 mm) as the reference point. The relative fishing intensity was estimated to be 0.20, 0.19, 0.18, 0.22, and 0.21 for the five pot types, indicating that an individual had a similar probability of entering any of the five pot types. The estimated relative size selectivity curves are shown in Fig. 4.

For the test pots, exploitation pattern indicators showed that the catch efficiency of undersized individuals significantly decreased with increasing escape vent size (Kruskal–Wallis test, $P < 0.01$). For instance, the pot with 70 mm × 20 mm escape vent retained 48.32% (*CI*: 45.33–51.68%) of individuals below the MLS ($nP-$); by comparison, $nP-$ would drop to 24.83% (*CI*: 21.34–28.28%) for the 70 mm × 25 mm escape vent, and less than 10% for the 70 mm × 30 mm and 70 mm × 35 mm escape vents, respectively (Table 3). However, the pots with large escape vent sizes might compromise decreasing catch efficiency for the legal-sized individuals. For the first three types of test pots (70 mm × 20 mm, 70 mm × 25 mm, and 70 mm × 30 mm), a slightly lower catch efficiency was observed for legal-sized individuals ($nP+ > 88\%$), while the catch efficiency of legal-sized individuals decreased significantly for the pot with 70 mm × 35 mm escape vent (Kruskal–Wallis test, $P < 0.01$), dropping by nearly 40% in the number. The additional indicators, $wP-$ and $wP+$, reflected a similar trend to $nP-$ and $nP+$. Of note, the $wP+$ of test pots (70 mm × 20 mm, 70 mm × 25 mm, and 70 mm × 30 mm) indicated no significant loss of catch efficiency for legal-sized individuals in the weight, as the *CI* contained 100% (Table 3).

Figure 5 shows the difference between the *CH*-dependent retention rates of different pot types. It is obvious that the increase in escape vent size significantly decreases the retention probability for undersized individuals, as the *CIs* of the four curves do not contain 0. For the first three delta plots, the difference in retention rates of undersized individuals

**Table 2  Estimated parameters and 95% confidence intervals (CIs) using the logistic model.**

|  | Control | 70 mm × 20 mm | 70 mm × 25 mm | 70 mm × 30 mm | 70 mm × 35 mm |
|---|---|---|---|---|---|
| CH50 | 10.73 | 18.23 | 23.19 | 25.73 | 31.79 |
|  | *10.45–10.91* | *17.98–18.43* | *22.93–23.85* | *25.02–26.25* | *30.80–32.52* |
| SR | 4.97 | 4.23 | 3.79 | 2.40 | 2.74 |
|  | *4.61–5.28* | *3.93–4.54* | *3.60–4.13* | *2.11–2.52* | *2.38–3.05* |
| P | 0.20 | 0.19 | 0.18 | 0.22 | 0.21 |
| Deviance | 0.84 | 0.71 | 0.97 | 1.26 | 2.03 |
| DOF | 8 | 8 | 8 | 5 | 5 |
| *P*-value | 0.99 | 0.99 | 0.99 | 0.94 | 0.84 |

**Note:**
CH50 is the carapace height with a 50% retention rate; SR is the selection range; P is the relative fishing intensity. DOF is the degree of freedom; *P*-value is the percent of deviance explained. CH50 and SR are in mm; The italics denote the 95% confidence intervals (CIs) of CH50 and SR.

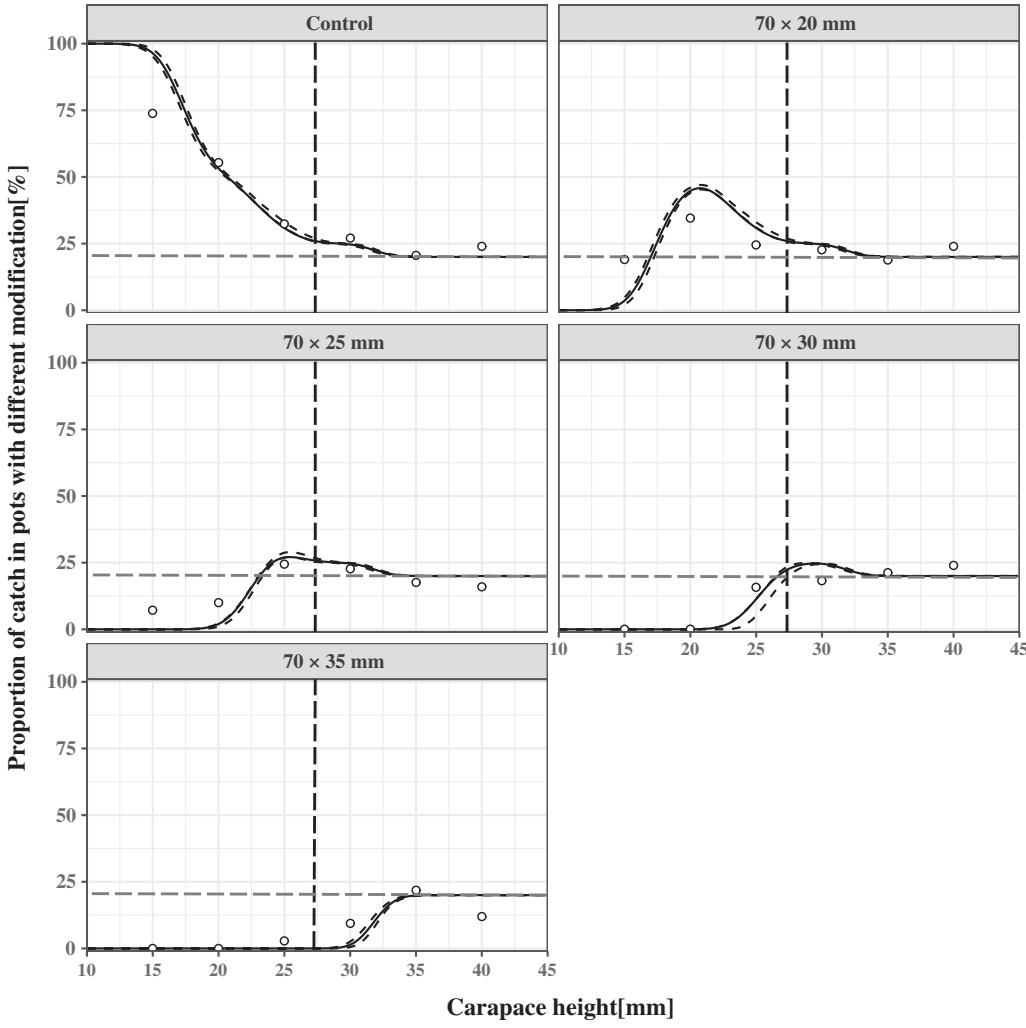

**Figure 4  Estimated relative size selectivity curves.** Thick solid curves denote the proportion of individuals caught in the five types of pots to the total catch. The dashed curves indicate the 95% confidence intervals for the fitted size selectivity curves. The hollow dots represent the experimental data. The horizontal gray baseline at 0.20 indicates equal catch efficiency, and the vertical line represents the MLS of carapace height (27 mm).     

**Table 3 Carapace height-based percentage and Efron 95% confidence intervals (CIs) of fractions below (nP−) and above (nP+) the MLS of carapace height (27 mm) in number of individuals and the ratio (nRatio) between nP− and nP+. Similar indicators based on weight (wP−, wP+, wRatio) have also been estimated. The italics denote the Efron 95% confidence intervals (CIs) of exploitation pattern indicators.**

|  | 70 mm × 20 mm | 70 mm × 25 mm | 70 mm × 30 mm | 70 mm × 35 mm |
|---|---|---|---|---|
| nP− | 48.32 | 24.83 | 5.37 | 2.68 |
|  | *45.33–51.68* | *21.34–28.28* | *3.95–6.16* | *0.00–5.52* |
| nP+ | 92.08 | 88.12 | 90.10 | 59.41 |
|  | *85.44–98.99* | *80.37–96.87* | *83.81–99.02* | *49.51–69.00* |
| nRatio | 0.52 | 0.28 | 0.06 | 0.05 |
|  | *0.46–0.61* | *0.22–0.35* | *0.04–0.07* | *0.00–0.12* |
| wP− | 54.31 | 26.95 | 6.79 | 5.36 |
|  | *49.74–59.31* | *22.16–31.44* | *4.40–9.40* | *0.00–10.63* |
| wP+ | 93.33 | 85.68 | 95.60 | 68.57 |
|  | *79.14–106.59* | *74.41–100.00* | *85.29–107.22* | *55.55–83.24* |
| wRatio | 0.58 | 0.31 | 0.07 | 0.08 |
|  | *0.47–0.73* | *0.24–0.41* | *0.04–0.10* | *0.00–0.17* |

gradually increases with increasing escape vent sizes and reaches the maximum values of 98%, 81%, and 54% in the *CH* classes of 14, 21, and 24 mm, respectively, while there is no difference in retention rates for legal-sized individuals. This result indicates that pots with 70 mm × 30 mm escape vents are the most efficient at releasing undersized individuals while maintaining catch efficiency for legal-sized individuals. For the last delta plots, the difference in the retention rates for legal-sized individuals is significant and reaches the maximum values of 99% in the *CH* class of 29 mm. This result shows that pot with 70 mm × 35 mm escape vent is less efficient at retaining the legal-sized individuals.

## DISCUSSION

This study presented novel results regarding the round pot selectivity and selection range of *C. japonica* from a small-scale fishery in the coastal waters of the Yellow Sea, China. Although several studies have investigated the performance of escape vents in pot fisheries (*e.g.*, *Boutson et al., 2009*; *Broadhurst, Butcher & Cullis, 2014*; *Broadhurst, Millar & Hughes, 2017, 2018*; *Broadhurst et al., 2019*), this study is the first to (1) systematically do research on the size selectivity of round pots for *C. japonica* in the coastal waters of the Yellow Sea, China, using a catch comparison method; (2) estimate the width of escape vents on the size selectivity of *C. japonica*; and (3) use a statistical approach to estimate individual fractions retained below and above the MLS of *CH* of *C. japonica*.

When crabs try to escape from the escape vents, they firstly side-crawl to the vents and then swim up and down to squeeze the body out the vents with the help of limbs and cheliped. *CH* limited the escape rate greatly according to the escape behaviour of *C. japonica*. Thus, we used *CH* instead of *CL* for determining size selectivity of escape vent (*Stasko, 1975*; *Brown, 1982*; *Treble, Millar & Walker, 1998*; *Rotherham et al., 2013*; *Broadhurst, Butcher & Cullis, 2014*; *Broadhurst, Butcher & Millar, 2017*). An MLS of

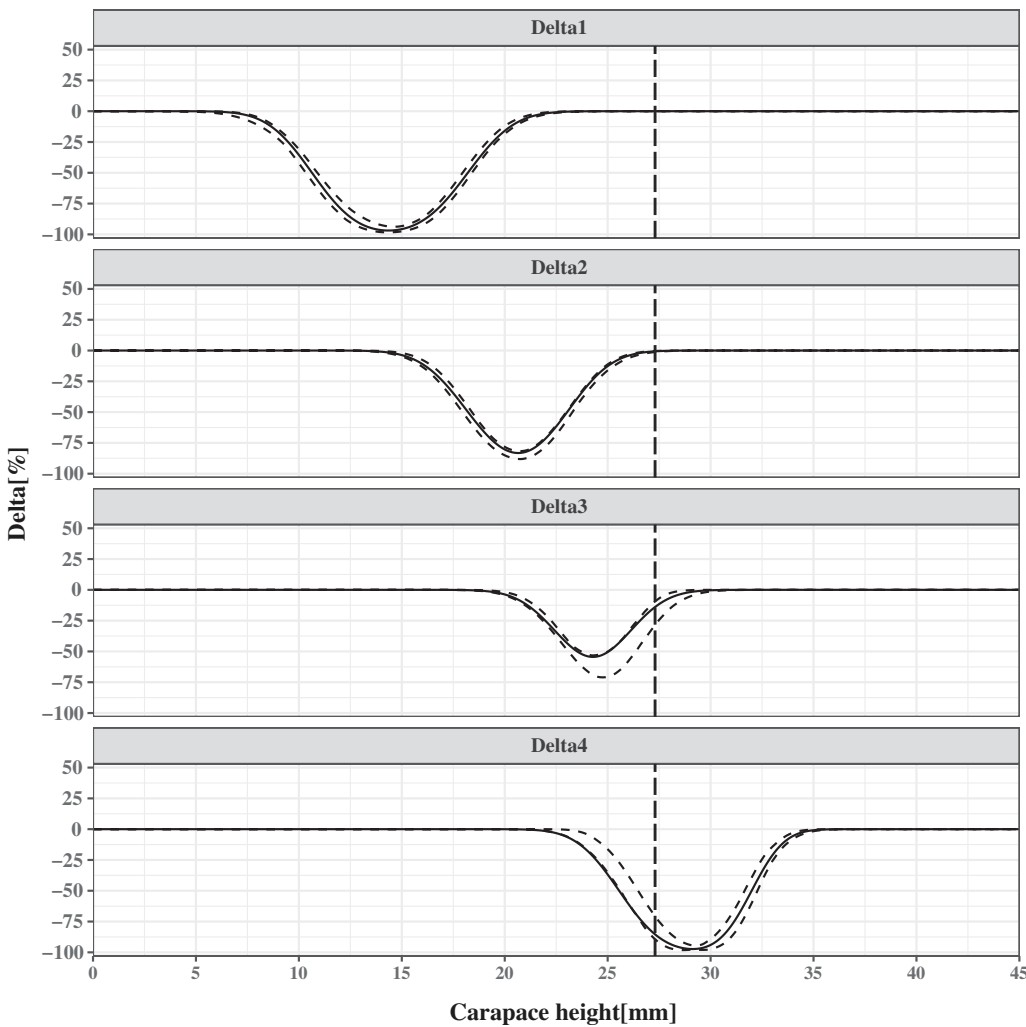

**Figure 5 Difference in the retention rates for pots between the escape vent sizes of control and 70 mm × 20 mm (Delta 1), 70 mm × 20 mm and 70 mm × 25 mm (Delta 2), 70 mm × 25 mm and 70 mm × 30 mm (Delta 3), and 70 mm × 30 mm and 70 mm × 35 mm (Delta 4).** Thick solid curves and dotted curves indicate the mean and 95% confidence intervals for the differences in fitted size selectivity curves, respectively. The vertical dashed line represents the MLS of carapace height (27 mm).

*CL* (50 mm) was formulated for *C. japonica* in China, but it does not discriminate between male and female populations, so we did not determine the males and females retained in the pots. The difference in mature length of male and female *C. japonica* implies sex ratio is valuable information in assessing the size selectivity of round pot for *C. japonica*, and this information will be accounted for in our future research (*Zhang et al., 2016*). The estimated MLS of *CH* was 27 mm based on the relationship between *CL* and *CH*, as crabs grow at the same rate in all linear dimensions (*Jirapunpipat et al., 2008*). The high correlation ($R^2$ = 0.98) between *CL* and *CH* indicates that we defined exploitation pattern indicators and explained the study results using the MLS of *CH* as the reference point was appropriate. The ideal gear will retain all legal-sized individuals and release all undersized individuals (*Guillory et al., 2004*). However, escape vent selectivity is

gradual, and crabs become less likely to escape with the increased size (*Guillory & Merrell, 1993*). Thus, the fractions retained in the pots below and above the MLS and the ratio between them can be good indicators of selective performance of fishing gear.

The results of this study indicated significant differences in size selectivity among pots with different-sized escape vents. The control pots presented poor selectivity, as the CH50 (10.73 mm) was well below the MLS of *CH* (27 mm), which will weaken the sustainability of fishery resources. With the installation of large escape vents, the CH50 increased considerably, implying that escape vents can effectively improve the size selectivity of round pots. This result is consistent with the published researches on the other commercially important crustacean pot fisheries (*e.g.*, *Guillory et al., 2004*; *Arana, Orellana & De Caso, 2011*; *Rotherham et al., 2013*; *Broadhurst, Millar & Hughes, 2017*; *Broadhurst, Butcher & Millar, 2017*; *Broadhurst & Millar, 2018a*; *Gandy et al., 2018*). The exploitation indicators indicated that the pots with 70 mm × 20 mm, 70 mm × 25 mm and 70 mm × 30 mm escape vents released nearly 50%, 75% and 95% of the undersized individuals, respectively, and these three types of pots retained approximately 90% of the legal-sized individuals compared with control pots without escape vents. The pot with 70 mm × 35 mm escape vent released all undersized individuals. However, they presented low catch efficiency for legal-sized individuals, as nearly 40% of these individuals were also released, similar to other studies (*e.g.*, *Rotherham et al., 2013*). Thus, more fishing effort is needed to compensate for the loss of legal-sized individuals. We recommend 70 mm × 30 mm as the most appropriate size for escape vents mounted on the round pots for *C. japonica* in the Yellow Sea of China following comprehensive consideration of local fishermen's profits and resource sustainability.

Several studies found that a reduction in undersized crab retention resulted in a higher catch of legal-sized crabs by avoiding the "pot saturation" phenomenon (*e.g.*, *Fogarty & Borden, 1980*; *Brown, 1982*; *Guillory & Merrell, 1993*; *Havens et al., 2009*; *Arana, Orellana & De Caso, 2011*; *Zhang et al., 2016*; *Broadhurst, Butcher & Millar, 2017*; *Zhang et al., 2020*). However, the number of legal-sized crabs was slightly lower in the test pots with 70 mm × 20 mm, 70 mm × 25 mm, and 70 mm × 30 mm escape vents than in the control pots without escape vents. This similar finding with the previous studies was probably caused by the low density of crabs in the fishing area (*Eldridge, Burrell & Steele, 1979*; *Guillory & Prejean, 1997*; *Treble, Millar & Walker, 1998*; *Guillory et al., 2004*; *Jirapunpipat et al., 2008*; *Boutson et al., 2009*; *Broadhurst & Millar, 2018a*), as we found that mean catch rates for legal-sized individuals were well below saturation thresholds. Species-specific behavioral interactions (*e.g.* competitive behaviors for food and space), size-specific distributions, and gear-specific factors (*e.g.* pot shape, volume, materials, entrance design and numbers) may also affect the entry and escape process when considering the passive capture characteristic of pot (*Kim & Ko, 1987*; *Yamane & Hiraishi, 2002*; *Vazquez Archdale et al., 2003, 2007*; *Montgomery, 2005*; *Vazquez Archdale & Kuwahara, 2005*; *Vazquez Archdale, Añasco & Hiromori, 2006*; *Butcher et al., 2012*; *Broadhurst, Butcher & Millar, 2017*).

Soak time proved to be an important factor that affects size selectivity and catch efficiency of pot gear (*Boutillier & Sloan, 1987*; *Montgomery, 2005*; *Nguyen et al., 2020*).

The retention of small-sized individuals can be reduced by increasing the soak time due to the long time for escaping (*Treble, Millar & Walker, 1998*; *Vazquez Archdale et al., 2007*; *Winger & Walsh, 2011*; *Olsen et al., 2019*). Bait attractiveness will decrease with time, and we found that bait was depleted when retrieving the pots during the sea trials. Thus, a soaking time of 2 days in our sea trials could provide sufficient chance for crabs to escape. Previous studies have shown that *C. japonica* could quickly detect the bait, then enter the pot to consume the bait, and final escape (*Vazquez Archdale, Kariyazono & Añasco, 2006*; *Vazquez Archdale et al., 2007*). Other species (by-catch), in addition to crabs, can also be trapped in the pots, and these species could have also fed on the bait. The availability of baits/lures could have affected the entry of crabs. This may also be a reason for the failure to detect a significant increase in the pots with escape vents in catchability for legal-sized individuals.

Ocean currents may cause discrepant catch effectiveness of round pots for undersized and legal-sized individuals because of the size-dependent swimming ability of crabs. However, *Vazquez Archdale et al. (2003)* found that most crabs (75%) crawled towards the pots downstream following the bait odor. This downstream swimming behaviour implies ocean currents will not cause quite different catch efficiency of traps for undersized and legal-sized individuals. Moreover, the entrances on the three sides of round pot provide much chance for undersized individuals to enter the gear. Further research using underwater video will provide more knowledge to verify this inference.

Escape openings are mandated in many crustacean fisheries around the world, such as Australian lobster fisheries and North American blue crab fishery (*Treble, Millar & Walker, 1998*; *Guillory et al., 2004*; *Broadhurst, Butcher & Millar, 2017*). The annual marine *Charybdis* landings have decreased from 62,581 to 24,259 tons (2010–2019) in China caused by environmental pollution, over-use of poor-selective fishing gears, and excessive fishing efforts, and this has raised immediate concerns of the Chinese government on restoring the coastal eco-system. Configuring a suitable escape vent was regarded as a simple and effective solution to conserve fisheries resources, mitigate ghost fishing mortality and achieve a balanced harvest in pot fisheries (*Boutson et al., 2009*; *Arana, Orellana & De Caso, 2011*; *Uhlmann & Broadhurst, 2015*; *Broadhurst & Millar, 2018b*; *Gandy et al., 2018*). The widespread use of pots in the coastal areas of China means that even marginal improvements in selectivity are likely to have considerable cumulative environmental and ecological benefits.

## CONCLUSION

In this research, we configured the escape vents on the round pots to improve the size selectivity of *C. japonica* in the coastal waters of the Yellow Sea, China. The catch comparison trials were conducted in the sea and the selective performances of four different escape vent width sizes (20, 25, 30, and 35 mm) with a fixed escape vent length of 70 mm were estimated. Five candidate selectivity models were compared and logistic model was selected as the best to describe the selectivity curves. The currently used pots are almost nonselective because of small mesh sizes, and configuring escape vents can significantly improve the size selectivity of *C. japonica*. With comprehensive analysis of the

retention rate of legal-sized individuals and the escape rate of undersized individuals, pots with an escape vent size of 70 mm × 30 mm were recommended considering local fishermen's profits and resource sustainability. This study can serve as a reference to develop more detailed fishery management regulations in the coastal waters of the Yellow Sea, China.

## ACKNOWLEDGEMENTS

We thank the Rizhao Fisheries Group Company for providing assistance, and we express our gratitude to Captain Teng for his advice and help during the sea trials.

### Funding

This work was supported by the Project of Marine and Fishery Technology Innovation of Shandong (No. 2017HYCX007). The funders had no role in study design, data collection and analysis, decision to publish, or preparation of the manuscript.

### Grant Disclosures

The following grant information was disclosed by the authors:
Marine and Fishery Technology Innovation of Shandong: 2017HYCX007.

### Competing Interests

The authors declare that they have no competing interests.

### Author Contributions

- Mengjie Yu conceived and designed the experiments, performed the experiments, analyzed the data, prepared figures and/or tables, authored or reviewed drafts of the paper, and approved the final draft.
- Liyou Zhang performed the experiments, prepared figures and/or tables, and approved the final draft.
- Changdong Liu analyzed the data, authored or reviewed drafts of the paper, and approved the final draft.
- Yanli Tang conceived and designed the experiments, authored or reviewed drafts of the paper, and approved the final draft.

### Data Availability

The raw catch data are available in the Supplementary File.

### Supplemental Information

Supplemental information for this article can be found online at http://dx.doi.org/10.7717/peerj.12282#supplemental-information.

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
