# Peer review of "Improving size selectivity of round pot for Charybdis japonica by configuring escape vents in the Yellow Sea, China"

_PeerJ, doi:10.7717/peerj.12282_

## Round 0.1 · original submission · Minor Revisions

The manuscript is well written and the experimental design is sound. The use of CH instead of CL is interesting. However, I would suggest the authors to discuss further on this based on the biology and behaviour of Charybdis japonica.

Specific comments:
1. Please make sure all abbreviations are used accordingly. For example, line 70, 'carapace height' should be replaced with 'CH'.
2. Figure 1, please convert the 'Vent size' into words.

Reviewer 1 ·

Basic reporting

In general, the manuscript was well written. However, improvements on the following are suggested :

● The title can still be reduced. It is too long.

●It is better to indicate the word crab in the first line under Abstract so that those who not familiar (especially non-crab workers/researchers) with Charybdis would immediately know.

●It is better to show the figure of the conventional crab pot design beside the modified round pot.

Experimental design

The experimental design was described well. However, information on the the basis/criteria in selecting the site for the deployment of crab pots should be included? How about the info on the crab population in the selected site?

Validity of the findings

In general, the data obtained were discussed well and the conclusions were based on the the acceptable discipline-specific repository.

There are some comments that need to be addressed though:

●Is there no MLS for male C. japonica in the decree of the Chinese Ministry of Agriculture? Is this the reason why the % males and females retained in the round pots with various sizes of escape vent was not determined? This could have been also a good information that can be useful in the future. In many fisheries ordinances, MLS for males and females are specified.

●Likewise, information on the by-catch was lacking (other animals that can be trapped in that type of crab pots? Animals other than crabs that entered the pots could have also fed on the bait. This could have affected the entry of crabs based on the availability of baits/lures (which serve as attractant).

Reviewer 2 ·

Basic reporting

This manuscript reports the result from catch comparison analysis of standard round pots vs pots with fitted escape vents to improve size selectivity. The set of experiments is well planned, the collection of data is appropriate and the analysis is valid. I find this manuscript suitable for publication after very minor revision.

Experimental design

The research question id clearly defines, relevant and meaningful. The set of experiments is well planned, the collection of data is appropriate and the analysis is valid. I would advise the authors to extent and clarify the definition of relative fishing intensity (lines 139-40). ..."relative to what"?

Validity of the findings

Definitely this study systematically assess and document the effect of different escape vents on the size selectivity of crabs in the Yellow Sea. This manuscript applies adequate models to carry out catch comparisons. It uses state of the art bootstrapping technique to estimate confidence intervals and applies exploitation indicators to assess the capture efficiency.

Additional comments

Minor comments:
Line 13: I would suggest deleting the word "little"
Line 139: It is a bit difficult to understand the term "relative fishing intensity". Please, provide a definition. relative to what? what do you mean by "intensity"

Reviewer 3 ·

Basic reporting

The author has data set, although I noted lack of details on the suitable analytical method for drawing conclusions. In addition, this manuscript is clearly written in a professional manner

Experimental design

Research question well defined

Validity of the findings

The findings on the effectiveness of window trap sizes are well presented well, with photograph and tables. However, there is a slight disturbance because I have not found out what method is used to conclude that there is a significant catch size between trap models with different window size
Still needed a clearer explanation about determining the best window trap size. Need a more suitable analytical method

Additional comments

Abstract:
It is necessary to add the analytical method used to determine the significance of each window traps size in selecting the size of the crabs caught

Introduction: good
Materials & method.
It is necessary to add the analytical method used to determine the significance of each window traps size in selecting the size of the crabs caught

Results: Crab catch information should be displayed in more detail
Preferably, the number of catches for each trip and each trap is displayed. Because if you look at the number of crabs caught, only 704 crabs (line 195). This number gives an indication that the crabs are very few. The number of traps is 50 with 9 hauls, so that when averaged only 1,56 crabs per trap for each haul.

Discussion: What about the influence of ocean currents on crab movement and the effectiveness of traps and the window in catching and selecting crabs according ti their size. This information will be very interesting and add value to this article

Conclusion: Good. The conclusion has answered the research objectives and the impact of the results of this research in the future

Annotated reviews are not available for download in order to protect the identity of reviewers who chose to remain anonymous.

---

## Round 0.2 · Minor Revisions

The authors addressed all our concerns in the current version of the manuscript. However, I would like to request the authors to please modify Figure 1A so that the dimension details are in the same background colour (white) as the common background. The current Figure 1A dimension is of poor quality.

---

## Round 0.3 · accepted · Accept

The authors have addressed all my concerns and those of the reviewers'. The design and analysis of this manuscript are in-depth and sound, and the results of the manuscript would contribute to the sustainable fishery of not only C. japonica, but other crab species in other geographical regions as well. I personally liked the clear and coherent writing of the manuscript! Well done.